# RST: Rough Set Transformer for Point Cloud Learning

**DOI:** 10.3390/s23229042

**Published:** 2023-11-08

**Authors:** Xinwei Sun, Kai Zeng

**Affiliations:** Faculty of Information Engineering and Automation, Kunming University of Science and Technology, Kunming 650500, China; sunxw@stu.kust.edu.cn

**Keywords:** 3D sensors, rough set, transformer, point cloud learning

## Abstract

Point cloud data generated by LiDAR sensors play a critical role in 3D sensing systems, with applications encompassing object classification, part segmentation, and point cloud recognition. Leveraging the global learning capacity of dot product attention, transformers have recently exhibited outstanding performance in point cloud learning tasks. Nevertheless, existing transformer models inadequately address the challenges posed by uncertainty features in point clouds, which can introduce errors in the dot product attention mechanism. In response to this, our study introduces a novel global guidance approach to tolerate uncertainty and provide a more reliable guidance. We redefine the granulation and lower-approximation operators based on neighborhood rough set theory. Furthermore, we introduce a rough set-based attention mechanism tailored for point cloud data and present the rough set transformer (RST) network. Our approach utilizes granulation concepts derived from token clusters, enabling us to explore relationships between concepts from an approximation perspective, rather than relying on specific dot product functions. Empirically, our work represents the pioneering fusion of rough set theory and transformer networks for point cloud learning. Our experimental results, including point cloud classification and segmentation tasks, demonstrate the superior performance of our method. Our method establishes concepts based on granulation generated from clusters of tokens. Subsequently, relationships between concepts can be explored from an approximation perspective, instead of relying on specific dot product or addition functions. Empirically, our work represents the pioneering fusion of rough set theory and transformer networks for point cloud learning. Our experimental results, including point cloud classification and segmentation tasks, demonstrate the superior performance of our method.

## 1. Introduction

Due to the significant advancements in autonomous driving and unmanned systems, LiDAR has emerged as a crucial component of three-dimensional perception systems. A point cloud comprises data points generated by 3D sensors, providing richer perceptual and interactive information compared to 2D scenes. Geometric information within point clouds forms the basis of numerous applications [1]. Three-dimensional sensors, such as LiDAR, generate irregular and unstructured reflective points that correspond to object surfaces [2]. Consequently, unlike 2D images, the disorderly and unstructured nature of 3D point clouds presents challenges in swiftly capturing semantic information directly from this data type [3].

Conventional point cloud learning methods [4,5,6] typically involve reordering the input point sequence or voxelating the point cloud to achieve canonical domain convolution. Nevertheless, as convolution-based models primarily focus on local structures, the disorder and irregularity of point cloud data significantly impact fine-grained localization accuracy. Recently, transformer-based models have garnered significant attention due to their superior ability to capture long-range dependencies. The core component for generating refined attention features based on global context is the self-attention module. Transformer-based models have addressed the limitations of convolution-based models and demonstrated significant advantages and potential in point cloud learning.

Owing to the disorder and lack of structure in point cloud data, there is a significant amount of uncertainty in the features, particularly evident at boundary points. Nevertheless, existing transformer models have not taken into account the potential misdirection and confusion caused by characterizing uncertainty in the attention mechanism. The self-attention mechanism relies on token-to-token relationships to guide the features directly through dot products or summation. This deterministic guided representation assumes that the features are correct and valid. Once there is a sudden occurrence of uncertainty token, the negative effect of this incorrect relationship is amplified in the global guidance. The current point cloud transformer clearly uses these deterministic guided representation methods to handle uncertain point cloud features, leading to a contradictory situation that is inappropriate. In fact, point clouds are not isolated; they share common features. It is feasible to create concepts by grouping similar point clouds and allowing these concepts to approximate each other to guide the representation, as depicted in Figure 1. However, there is insufficient research on more objective guidance mechanisms to accommodate the disorder and lack of structure in point cloud data. Therefore, it is essential to introduce a novel feature representation method to handle uncertain tokens.

In recent years, rough set theory has been considered as a powerful mathematical tool for the uncertainty issue [7]. Rough set theory forms concepts by the granulation of a cluster of tokens, then expresses the similarities and differences between concepts via an approximation operator. It can extract the most important features to achieve its concept induction [8]. Obviously, rough set is an effective method for studying the characterization of uncertain conceptual relationships. It is more capable of guiding the relationship between global information. Nevertheless, we empirically found that none of the current transformers constructed based on rough set methods improve the feature extraction capability of the model. The key reason is that the most popular existing rough set models are used for the consistency measurement of the issue of feature selection. In these research works, the samples are described by both conditional and decisional attributes. For our study, point cloud data are in the same feature space. It necessitates redesigning the process of approximation computation.

To tackle these problems, an attention mechanism grounded in rough set theory is proposed to accommodate the presence of uncertain tokens during the guidance process. The granulation and lower-approximation expressions are redefined for neighborhood rough sets to align with the fundamental rough set definition, thus enhancing their suitability for deep learning applications. The technique leverages neighborhood granulation to naturalize point cloud features, forming information granules. Feature guidance is subsequently achieved by quantifying the similarity between information granules through lower approximation. Unlike the traditional dot product attention in transformers, this approach demonstrates an improved resilience to the inherent uncertainty within point cloud data, resulting in a more objective representation.

As a result, it effectively addresses the limitations of the current point cloud transformer.

The major contributions are as follows:1We redefine the granulation and lower-approximation expressions for neighborhood rough set to conform to the fundamental definition of rough sets and enhance their applicability in deep learning. Through empirical investigation, we have determined that this marks the initial fusion of rough set theory and deep learning network models in the context of point cloud learning.2We propose a novel rough set-based attention mechanism to replace the dot product attention, thereby constructing a transformer network structure (RST) tailored for point cloud learning. This network directly takes point cloud data as inputs and extracts features using multi-head rough set attention. In comparison to the traditional transformer model, the RST network exhibits a stronger ability to provide an objective relationship guidance for uncertain point cloud data.3The model is evaluated through point cloud classification and segmentation experiments using the ModelNet40 [9] and ShapeNet [10] datasets. All the results demonstrate that our method outperforms the most advanced networks. Additionally, we conduct a visual analysis to elucidate the improvements over traditional attention mechanisms. The resource codes are validated at https://github.com/WinnieSunning/RST, (accessed on 7 November 2023).

## 2. Related Work

In this section, we will conduct an in-depth literature review of the state-of-the-art deep learning methods for 3D LiDAR point clouds, including traditional point cloud learning methods, transformer-based point cloud learning methods, and other advanced point cloud learning methods.

### 2.1. Traditional Point Cloud Learning Methods

Before the transformer method became popular, the deep learning methods on point clouds can be roughly categorized into two classes: PointNet-based [11,12,13] and convolution-based [4,5,14]. PointNet [11] uses the multi-layer perceptron (MLP) and max pooling operation to extract features of point clouds. PointNet can perform the classification task well, but the local features’ extraction ability is poor, which makes it difficult to analyze complex scenes. Inspired by CNN, PointNet++ [12] is proposed, which is able to extract local features at different scales and obtain deep features via a multi-layer network structure. In a recent study, Wijaya et al. proposed a new PointNet-based point cloud feature learning network: PointStack [15]. It features multi-resolution feature learning and learnable pooling, which further reduces the information loss during sampling. PointCNN [5] proposes a new convolution operator that converts input points and features to data on a regular grid and then applies convolution. Similarly, PointConv [6] treats convolution kernels as nonlinear functions of the local coordinates of points composed of weight and density functions.

The traditional approach has two main shortcomings. One is that its global or local information is easily lost, leading to poor generality and difficulty in adapting to complex tasks. Second, due to the disorderly nature of point clouds, the use of convolution methods must consider transforming their inputs [16], which leads to further information loss as well as computational cost.

### 2.2. Transformer-Based Point Cloud Learning Methods

The transformer-based methods are mainly divided into: local transformer and global transformer. Local transformers are designed to implement feature aggregation in local patches rather than the whole point cloud. PT [17] adopted the PointNet++ hierarchical architecture for point cloud classification and segmentation. It focused on local patch processing, and replaced the shared MLP modules in PointNet++ with local transformer blocks. Compared to the local transformer, the global transformer can connect each output feature to all input features and can learn global contextual features. PCT [18], as a pure global transformer network, was proposed in. PCT takes 3D coordinates as input, and employs a neighborhood-embedding structure to map the point cloud to a high-dimensional feature space. 3CROSSNet [19] is the first transformer network that can extract features across scales in point cloud learning. It consists of three components, which can perform self-attention operations on point clouds at different levels and scales, respectively, to extract rich features.

The existing transformer models above consider the point as an isolated element, which can be described as a feature vector. Then, the dot product or addition relations between two points are employed for guiding the generation of feature maps. As a matter of fact, these unordered points are not standalone, and they form a meaningful subset, which can be regarded as certain concepts. We should seek relationships between concepts in their commonalities to characterize them via approximate methods rather than specifying the dot product or addition function.

### 2.3. Other Advanced Point Cloud Learning Methods

In addition to research focusing on modifications to the overall feature extraction network, several state-of-the-art deep learning methods have been applied in practical 3D LiDAR point cloud applications. Li [20] examined a rapid deterministic point set registration method that utilizes gravity prior information, improving the speed and accuracy of point cloud registration. A self-attention and orientation coding network was constructed by Yan et al. [21], which integrates long-distance context into point local descriptors, fully explores the relationship between points, and solves the problem of position recognition of point cloud data. Wu and Tian et al. [22] proposed the Point Prompt Training framework to support 3D representation learning for multi-dataset collaborative learning and overcome the negative transfer related to collaborative learning. Zhu and Yang [23] introduced a comprehensive 3D pre-training framework aimed at enabling the acquisition of efficient 3D representations through the learning of point cloud representations via differentiable neural rendering.

Aligning with recent research trends in point cloud registration, scene recognition, large-scale 3D representation learning, and general pretraining paradigms. It is evident that fundamental theoretical disciplines like physics and mathematics offer substantial theoretical support and technical inspiration for deep learning methods. Therefore, addressing the challenges in existing point cloud learning through mathematical approaches holds significant research value and significance.

## 3. Method

In this section, we will begin by introducing the redefinition of granulation and lower-approximation operators to make them applicable to deep learning networks. Building upon this, a rough set-based attention mechanism is developed. Subsequently, based on this foundation, we construct a transformer network for point cloud learning.

### 3.1. The Rough Set-Based Attention Mechanism

For the neighborhood rough set that we employ, the mutually exclusive information granules generated by neighborhood relations serve as the fundamental units for approximation. Below, we first introduce the fundamental properties of the neighborhood relations adopted.

**Definition 1 ([24]).** 
*Let U be a non-empty set, if, for any element $x_{i},x_{j},x_{k}$ in U, there exists a uniquely determined real function *Δ* corresponding to it, and *Δ* satisfies:*

*(1) $\Delta (x_{i},x_{j})\geq$ 0. When and only when $x_{i}=x_{j}$, $\Delta (x_{i},x_{j})=$ 0.*

*(2) $\Delta \left(x_{i},x_{j}\right)=\Delta \left(x_{j},x_{i}\right).$*

*(3) $\Delta \left(x_{i},x_{k}\right)\leq \Delta \left(x_{i},x_{j}\right)+\Delta \left(x_{j},x_{k}\right).$*

*where *Δ* is a distance function on U and <U,Δ> is a distance space, also called a metric space.*

*In the N-dimensional Euclidean space, given any two points $x_{i}=\left(x_{1i},x_{2i},\ldots ,x_{Ni}\right)$ and $x_{j}=\left(x_{1j},x_{2j},\ldots ,x_{Nj}\right)$, the distances are:*

(1)
$$\Delta \left(x_{i},x_{k}\right)={\left(\sum_{l=1}^{N}{\left(x_{li}-x_{lj}\right)}^{2}\right)}^{\frac{1}{2}}$$



The specific definition of the distance function can have various expressions due to the existence of multiple variables [25], but they all play the role of measuring the distance relationship between any two points in the Euclidean space.

Information granules generated through neighborhood relations can be further guided through upper and lower approximations. Below, we provide the definition of upper and lower approximations employed in our context.

**Definition 2 ([24]).** 
*Given a nonempty finite set $U=\left\{x_{1},x_{2},\cdots ,x_{n}\right\}$ on a real space and a neighborhood relation on U domain relation N, we call the NAS = <U,N> as a neighborhood approximation space.*


**Definition 3 ([24]).** 
*Given NAS = <U,N> and X ⊆ U, X is in the neighborhood approximation space, the lower and upper approximations of NAS = <U,N> are defined as:*

(2)
$$\underline{N}X=\left\{x_{i}|\delta \left(x_{i}\right)\subseteq X,x_{i}\in U\right\}$$


(3)
$$\bar{N}X=\left\{x_{i}|\delta \left(x_{i}\right)\cap X\neq ⌀,x_{i}\in U\right\}$$


*It has the following properties:*

(4)
$$\forall X\subseteq U,\bar{N}X⊇X⊇NX$$



In the neighborhood rough set, mutually exclusive information particles generated by neighborhood relations are the basic units used for approximation. It can be said that the two modules of granulation and approximation form the rough set methodology’s cornerstone. Therefore, we will design the self-attention mechanism based on the two modules of granulation and approximation.

The following describes the granulation module. The fuzzy equivalence relation functions *R* used to measure between samples are diverse. To better extract the commonality among the high-dimensional features of the point cloud data, we generated the relational functions for the granulation operation with Gaussian functions.
(5)$$R\left(x,y\right)=exp\left(-\frac{||x-{y||}^{2}}{2\sigma^{2}}\right)$$

The Gaussian function satisfies all the properties needed to be defined as a rough set relation function for n feature vectors X of length m.

The granulation matrix has the following properties:

(1) ∀*x*,*y*∈*U*, R(x,x) = 1;

(2) R(x,y)= R(y,x);

(3) R(x,y)∈ [0,1].

The granulation matrix serves as an information bottleneck in this process. It reflects the relationships between objects, expresses the granular structure of the argument domain, and carries all the sample information available to the approximation operation. Essentially, once the relationships between samples are extracted, subsequent rough calculations are performed on the fuzzy information grains constituted between samples rather than on individual samples.

The following describes the approximation module. In the fuzzy calculation of rough sets, the classification of samples is no longer either 0 or 1 in the deterministic sense, but is evaluated in terms of fuzzy affiliation between 0 and 1. For all information grains generated by the relational function, the mutual approximation between different fuzzy information grains can guide the importance of each other, and this property makes the approximation matrix a natural global feature guidance matrix, while the affiliation of the approximation is reflected by the upper and lower approximations of the rough set.

For the relational function k(x,y), the lower and upper approximation affiliations to the approximated information grain $d_{i}$ are:(6)$$\underline{k_{S}}d_{i}\left(x\right)=\underset{y\in U}{\inf}S\left(N\left(k\left(x,y\right)\right),d_{i}\left(y\right)\right)$$
(7)$$\bar{k_{T}}d_{i}\left(x\right)=\underset{y\in U}{\sup}T\left(k\left(x,y\right),d_{i}\left(y\right)\right)$$

Among them, where *N* is the complementary operator, *T* is the trigonometric norm and *S* is the trigonometric remainder.

By approximating the different concepts ($d_{i}$) formed through granulation to each other, the fuzzy lower-approximation affiliations of ∀*x* ∈*U* affiliation to d through the relational function *R*, which we construct, can be expressed as:(8)$$\;\underline{k_{S}}d_{i}\left(x\right)=\underset{y\notin d_{i}}{\inf}\;\min\left(1\;-\exp\left(-\frac{{∥x-y∥}^{2}}{2\sigma^{2}}\right),\;R\left(x,y\right)\right)$$

The value of the lower approximation characterizes the necessary correlation affiliation between the information grains composed of two features with each other. The granulation operation and approximation operation are calculated as shown in Figure 2.

The approximation operation more objectively and accurately measures the degree of correlation between two information grains, which leads to the correlation of global features.

The rough set-based attention mechanism constructed based on granulation and approximation operations, where $\left(Q,K,V\right)\in R^{N\times D}$ is generated by shared linear transformations and the input features F. The specific calculation of the rough set-based attention mechanism is as follows.

First, we can use the query and key matrices to calculate the granulation matrix (*G*) by *R*:(9)$$G=R(Q,K^{T})$$

The granularity matrix is naturally normalized due to the design of the relationship function.

Then, *G* regenerates a weight matrix *A* of the same size as the granulation matrix via the approximation operation, whose values characterize the degree of necessary correlation between information grains and are used as the weights of attention. The rough set-based attention mechanism output features $F_{sa}$ are the weighted sums of the value vector using the corresponding attention weights:(10)$$F_{sa}=A\cdot V$$

The whole rough set self-attention computation process is carried out in the form of information grains, and both its granulation and weighted sum are permutation-independent operators. Therefore, the rough set-based attention mechanism is better adapted to the irregularity and disorder of the point cloud, while the correlation between the features is better measured. The overall approximate guided representation of rough set attention is shown in Algorithm 1.
**Algorithm 1** Approximate guided representation methods based on rough set1:A feature matrix $Q=\left\{X_{1},\cdots ,X_{d}\right\}$(d tokens of length n)2:Granulation operation:3:**for** each token X **do**4:   $R=\left(X_{i},X_{j}\right)$ Calculate each token with the others by Formula (6);5:**end for**6:Obtain the granulation matrix $G=\left\{R_{11},R_{12},\cdots ,R_{nn}\right\}$ (n information granule g of length n)7:Approximation operation:8:**for** each information granule g **do**9:   $\underline{k_{S}d_{ij}}=g_{i}\leftarrow g_{j}=\inf\left(\left(1-R_{ij}\right)\wedge R_{ij^{\prime }\left(j^{\prime }\neq j\right)}\right)$ The lower approximation is calculated by Formula (9);10:**end for**11:Obtain the weight matrix $A=\left\{\underline{k_{S}d_{11}},\underline{k_{S}d_{12}},\cdots ,\underline{k_{S}d_{mn}}\right\}$12:return A.

### 3.2. Transformer Network for Point Cloud Learning

The overall network structure is shown in Figure 3. The transformer network based on rough set attention consists of three components. In the Input Embedding module, the coordinate information and features of the point cloud are directly input to the high dimension, which is used to encode the coordinates and features of each point. We mainly refer to PCT for the design here, given an input point cloud $\rho \in R^{N\times d}$ where each *N* point has a *d*-dimensional feature description, a $d_{e}$- dimensional embedded feature $F_{e}\in R^{N\times d_{e}}$ is first learned via the Input Embedding module. The embedded features are then entered into the encode module. And in encoder, multiple rough set-based multi-headed attentions constitute the main forward network for feature extraction. The output of each attention layer is aligned with the dimensionality of the input. Finally, the output of the attention layer is fed through multiple linear layers to the subsequent classification decoder. The architecture of the segmentation network decoder is almost the same as that for the classification network, following most other point cloud segmentation networks. The loss function uses the softmax cross entropy loss.

The rough set attention completes the feature bootstrapping of the global information and extracts the main features. Finally, classification and segmentation are performed by the decoder. Our transformer is framework-based, and it can implement a wide variety of point cloud learning tasks, with a modular self-attention to allow for the direct replacement of traditional self-attention.

## 4. Experiments

We discuss the performance of the rough set transformer net for two classical point cloud learning tasks: shape classification and part segmentation. The experiments were run on a server equipped with a Tesla V100 GPUs, and the operating system was Ubuntu 20.04.

### 4.1. Classification on ModelNet40

In this section, we have completed cloud classification experiments on the ModelNet40 dataset, which comprises 40 categories of 3D models. The ModelNet40 dataset includes 9843 training samples and 2468 testing samples. The same sampling strategy as used in PCT was adopted to uniformly sample each object to 1024 points. For the classification task, we employed two evaluation metrics: overall accuracy (OA) and mean accuracy (mA). We also compared the most representative models based on non-transformer and transformer approaches. During training, we utilized random translation, random anisotropic scaling, and random input dropout strategies to augment the input points’ data during training. During testing, no data augmentation or voting methods were used. The SGD optimizer was used for 400 epochs with the batch size 32. We set the initial learning rate to 0.0001 and adopted a cosine annealing schedule to adjust the learning rate at every epoch.

The experimental results are shown in Table 1. It is apparent that our method outperforms most previous models, especially the transformer-based models. Firstly, compared to the original transformer, we conducted two sets of experiments based on addition and dot product relationships. It can be observed from the experimental results that our method achieved a 1.3% and 0.9% improvement in mA, as well as a 2.8% and 1.8% improvement in OA. Compared to the baseline model PCT, our model makes a 0.5% improvement, respectively, on overall ACC. Additionally, it achieves the result of 90.8% mean accuracy, which is 0.8% higher than the PCT. The experimental results visually verify the success of our design of a transformer model based on rough set theory. Most point input-based models, such as the PointNet family, have only 92% OA results and perform even worse on mA. The transformer-based methods are generally better than the non-transformer-based methods, generally achieving a 93% OA. This is due to the transformer’s ability to learn global information. There are also a few methods, such as PointConv and Point Transformer, which rely on increasing the number of inputs or fusing voxels to improve accuracy. Compared to these, our model achieves the best performance in terms of OA and mA, while still having great potential for improvement.

In order to emphasize the advantages of the neighborhood granulation we employed, the experimental results for other conventional rough set granulation methods are shown in Table 2. Obviously, the overall performance of the rough set-based transformer model is significantly superior to the traditional transformer. It is especially reflected in the mA metric, which is attributed to the tolerance of the rough set approximation-guided representation to the uncertain information. It is worth clarifying that different granularization relations will be the most appropriate choice depending on the characteristics of different data. Since the attention of the point cloud has a strong influence on the neighboring points, the granulation function of the Gaussian kernel function obtains the best performance.

In Table 3, the degree of fit of the model to the data is evaluated, reflecting the accuracy of the model in classifying point cloud data. From the results, it can be observed that our method achieves the lowest error, which further validates the superiority of the method.

The disordered and unstructured point clouds make it challenging to quickly capture semantics information directly based on local models such as CNNs. The transformer-based models are competent because of its superior capability in capturing long-range dependencies. However, the non-local method is still inadequate for the complex point cloud data. Our rough set attention forms the concepts of features via the granulation of a cluster of tokens rather than an isolated feature vector, which is more objective than the recent transformer model. Then, it expresses the similarities and differences between concepts through a lower-approximation operator. The experimental results show that our method is more suitable for the uncertain characterization of point cloud classification.

### 4.2. Part Segmentation on ShapeNet

The model is also evaluated on ShapeNet dataset for part segmentation. ShapeNet consists of a total of 16,880 models, 14,006 of which are used for training and the rest for testing. In our experiment, we directly used the same train–test segmentation as PCT. 2048 points were sampled from each model as input, and only a few point sets had six parts with labels. In the segmentation task, we used mean Intersection-over-Union (mIoU) as an assessment metric. We selected representative networks using point input and self-attention as the main feature extraction for comparison. A training setup essentially identical to our classification task was used. We train our model using the SGD optimizer for 500 epochs with the batch size 16. The initial learning rate was set to 0.0001, with a cosine annealing schedule to adjust the learning rate at every epoch.

The experimental results are shown in Table 4. Similarly, the transformer-based models have obtained very good results. Models that use self-attention and other variants are primarily compared. In order to highlight the impact of our attention, we did not add any other modules and still obtained the best performance. Comparing the baseline models NPCT and SPCT for the same training case. The NPCT and SPCT underperformed us by 0.8 and 0.2, respectively. For the classical PointNet family, we obtained a boost of 0.9–2.3 mIoU. Our rough set-based self-attention achieves the best performance when using only the self-attention and transformer framework.

It is certain that our transformer yielded better performances because of the non-local feature extraction. Nevertheless, the existing transformer models consider the point as an isolated element, which can be described as a feature vector. Then, the dot product or addition relations between two points are employed to guide the generation of feature maps. On the contrary, our granular relation is able to describe the commonalities among multiple points. Then, we characterize their commonalities via approximate methods rather than specifying the dot product or addition function. This is the key reason as to why we have a better mIoU result.

Efficiency is a topic of concern. To this end, we add a comparative experiment between the traditional transformer model (PCT) and the rough set-based transformer in terms of efficiency. Herein, we display our time tests on the RTX2060 with a batch size of 16, shown in Table 5.

Clearly, the rough set-based transformer is more time-consuming than the traditional transformer (dot product attention) due to its intricate algorithm design and additional computations. Nevertheless, we have optimized it to reduce the time required, incorporating algorithm enhancements and developing CUDA parallel computing libraries. At present, the time required by our method is nearly equivalent to that of the traditional transformer. We have open-sourced the code and library files for this approach as part of our contribution. As a result, our approach offers a more comprehensive performance improvement, leading to a superior overall efficiency compared to the traditional transformer.

### 4.3. Visualization Analysis Experiments

To further analyze the improvements of our model, we visualized the weight matrix after its approximation operation. As shown in Figure 4, we performed one shallow-layer and one deep-layer visualization on the ModelNet40 and ShapeNet datasets. The rough set attention and dot product attention are employed for the comparison analysis.

The feature maps both on the shallow layer and the deep layer are all more obviously brighter than dot product attention. This demonstrates that rough set attention has a stronger global information extraction capability than dot product attention [38]. The traditional attention mechanism further suffers from information loss in the deep network. The disordered and unstructured point clouds result in uncertainty which the traditional attention is incapable of handling. On the contrary, rough set attention can deal with this uncertainty well by measuring the similarity in the view of approximation. It greatly reduces the effect via uncertain point cloud data.

To further validate the rough set-based transformer model’s ability to handle uncertainty information, we visualize the classification effect of a set of point cloud data with random unordered inputs. The results are shown in Figure 5, whereby the input comprises 1024 point cloud data, totaling 40 groups of different types of labels. The distance between the point clouds represents the degree of model recognition: the more obvious dispersion indicates a better recognition effect. The upper part shows the traditional transformer method; it can be seen that our method obviously has a better classification effect on uncertainty features than the traditional method.

Occlusion is also a common challenge that point cloud data often encounter in real-world scenarios. To validate our effective handling of such uncertainty features, we devised three levels (10%, 30%, 50%) of feature masking to assess the model’s learning performance, and the results of the comparative experiments with the traditional transformer are presented in Figure 6. Upon analyzing the experimental results, it becomes evident that as occlusion intensifies, the performance of the dot product attention-based transformer significantly deteriorates, whereas RST displays robustness in handling occluded features. This is attributed to the rough set-based attention mechanism’s ability to tolerate occluded tokens, resulting in a more objective guidance.

After experimental comparative analysis, we verify that the lower-approximation operator is representational, and also prove that it is advantageous compared to existing operators such as multiplicative operators. The reason for this analysis can be attributed to the fact that the approximation operator is able to characterize the uncertain and inconsistent points or problems inside the point cloud, and has tolerance for these uncertain relations, so it is superior compared to the multiplicative relations.

## 5. Conclusions

Recently, transformer models have exhibited excellent performance in addressing the challenge of point cloud learning. Nonetheless, the disorder and lack of structure in point clouds introduce uncertainty, which needs to be addressed within the traditional transformer framework. In this study, we introduce a novel rough set attention mechanism and extensively investigate the rough set transformer (RST) network. It is able to accurately classify and segment the point cloud data collected using LiDAR. RST has a more objective relationship guidance ability for the uncertain point cloud. All the experimental results have shown that our method yielded a better performance than the most advanced networks available.

## Figures and Tables

**Figure 1 sensors-23-09042-f001:**
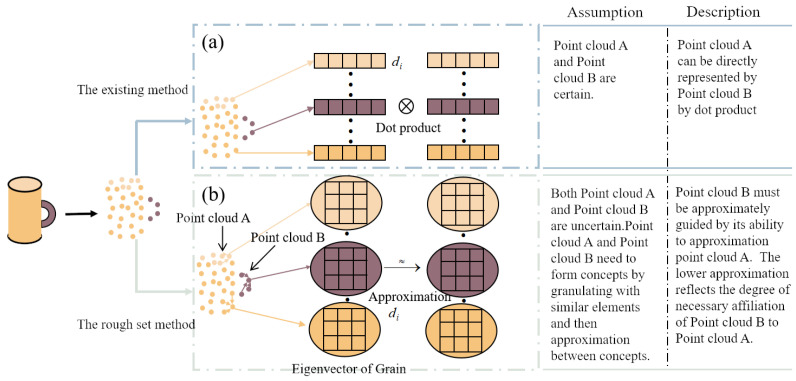
The relation representation of point cloud data. (**a**) The dot product or addition relations between two isolated feature vectors in classical transformer. (**b**) The approximation relations after granulation in the view of rough set theory.

**Figure 2 sensors-23-09042-f002:**
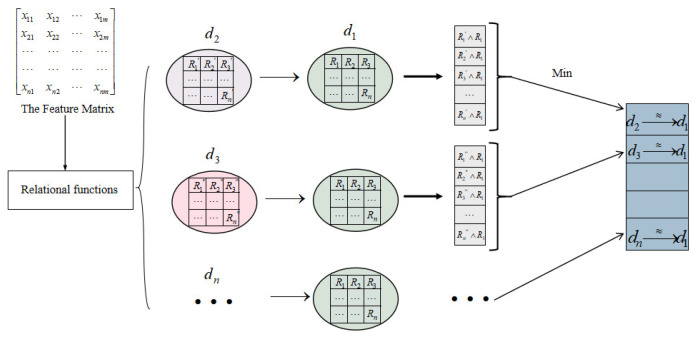
The granulation operation and approximation operation. The relationship function guides the commonality between each feature, and the generated granulation matrix contains all the information about each feature with other features. The mutual approximation between concepts leads to the degree to which information grains are necessarily related to each other.

**Figure 3 sensors-23-09042-f003:**
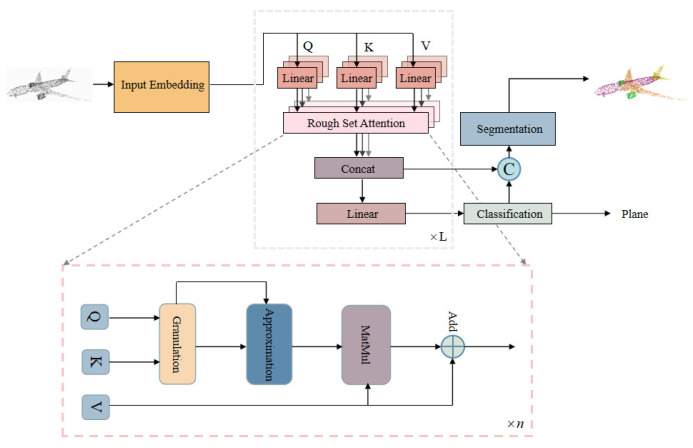
Architecture of RST. The encoder mainly comprises an Input Embedding module and multiple multi-headed attention modules. The decoder mainly comprises multiple linear layers for classification and segmentation.

**Figure 4 sensors-23-09042-f004:**
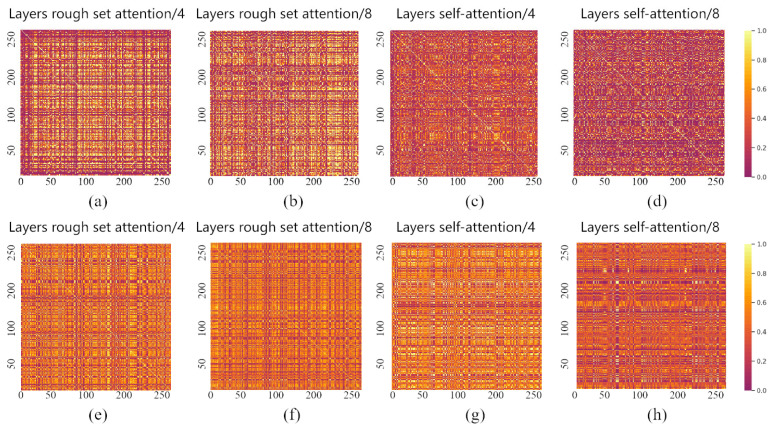
Attention visualization on the ModelNet40 and ShapeNet dataset. (**a**,**b**,**e**,**f**) Heat maps based on the rough set. (**c**,**d**,**g**,**h**) Heat maps based on the traditional transformer.

**Figure 5 sensors-23-09042-f005:**
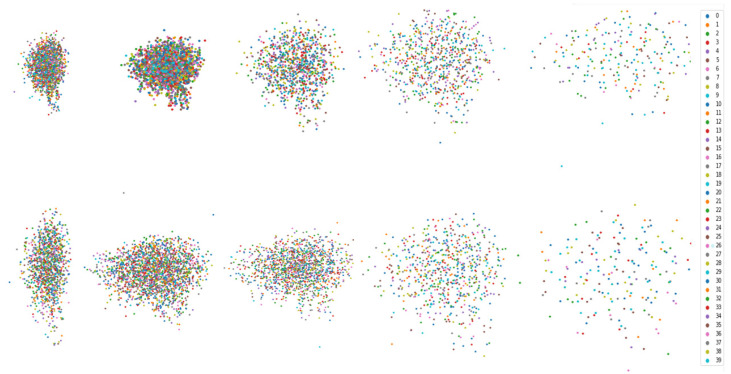
Visualization of classification results. The upper part shows the point cloud classification effect of common transformer models, and the lower part shows the effect of a rough set-based transformer model. The better the dispersion effect, the better the classification result.

**Figure 6 sensors-23-09042-f006:**
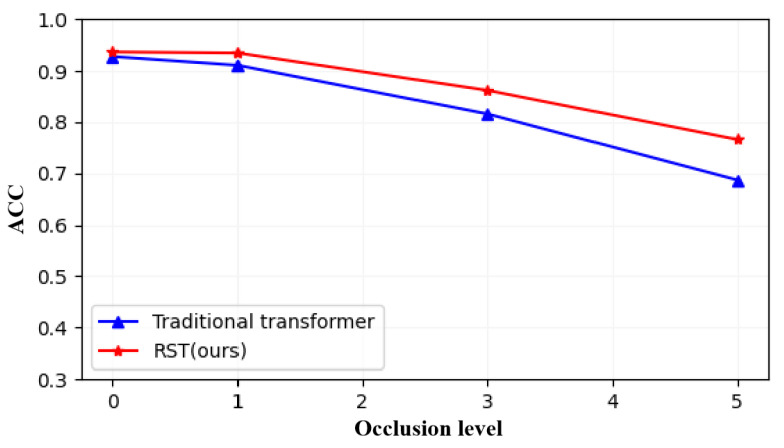
Comparison experiments between traditional transformer and RST with different occlusion levels.

**Table 1 sensors-23-09042-t001:** Comparison on the ModelNet40 classification dataset. * represents the baseline.

Method	Input	Input Size	OA (%)	mA (%)
PointNet [11]	P	1024 × 3	89.2	86.2
PointNet++ [12]	P,N	5120 × 6	91.9	–
PlaneNet [13]	P	1024 × 3	92.1	90.5
PointConv [5]	P,N	1024 × 6	92.5	88.1
PointCNN [6]	P	1024 × 3	92.2	88.1
DGCNN [14]	P	2048 × 6	93.5	90.7
Point2Seq [26]	P	1024 × 3	92.2	90.4
RSMix [27]	P	1024 × 3	93.5	–
Manifold [28]	P	2048 × 6	93.0	90.4
PointStack [15]	P	1024 × 3	93.3	89.6
DGCNN+MD [29]	P	1024 × 3	93.3	89.99
OGNet+MD [29]	P	1024 × 3	93.39	90.71
PointASNL [30]	P,N	1024 × 6	93.31	–
Add-attention	P	1024 × 3	92.4	88.0
Bmm-attention	P	1024 × 3	92.8	89.0
PCT [18] *	P	1024 × 3	93.2	90.0
3DMedPT [31]	P	1024 × 3	93.4	–
3CROSSNet [19]	P	1024 × 3	93.5	–
PatchFormer [32]	P,N	1024 × 6	93.6	–
PT [17]	P,N	1024 × 6	93.7	90.6
LCPFormer [33]	P	1024 × 3	93.6	90.7
RST (ours)	P	1024 × 3	93.7	90.8

**Table 2 sensors-23-09042-t002:** Comparative experiments with other granulation relations.

Granulation Relations	OA (%)	mA (%)
Dominant Relationship	93.0	90.0
Euclidean Norm	93.2	90.2
Multiquadric Kernel	93.6	90.4
Gaussian Kernel	93.8	90.8

**Table 3 sensors-23-09042-t003:** Normal estimation average cosine distance error.

Method	Points	Error (%)
PointNet [11]	1k	0.47
PointNet++ [12]	1k	0.29
PCNN [5]	1k	0.19
RS-CNN [34]	1k	0.15
PCT [18]	1k	0.13
RST (ours)	1k	0.11

**Table 4 sensors-23-09042-t004:** Comparison on the ShapeNet part segmentation dataset. * represents the baseline.

Method	mIoU (%)
PointNet [11]	83.7
3DMedPT [31]	84.3
ShapeContextNet [35]	84.6
PointNet++ [12]	85.1
P2Sequence [26]	85.1
DGCNN [14]	85.2
DT-Net [36]	85.6
PointConv [5]	85.7
3CROSSNet [19]	85.9
CAA [37]	85.9
PT [17]	85.9
NPCT [18] *	85.2
SPCT [18] *	85.8
PointCNN [6]	86.1
RS-CNN [34]	86.2
RST (ours)	86.5

**Table 5 sensors-23-09042-t005:** Comparison experiments between PCT and RST with efficiency.

Method	Epochs (Convergence)	Average Training Time	Average Testing Time
PCT	250	79.13 s/epoch	8.58 s
RST (ours)	300	79.58 s/epoch	8.67 s

## Data Availability

Data are contained within the article.

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
