# Peer review of "RST: Rough Set Transformer for Point Cloud Learning"

_sensors, 2023, doi:10.3390/s23229042_

Round 1

Reviewer 1 Report

Comments and Suggestions for Authors

This paper proposes a novel rough set attention mechanism for point cloud learning. Overall, the structure of this paper is well organized, and the presentation is relatively clear. However, there are still some crucial problems that need to be carefully addressed before a possible publication.

1.  Please give more details and discussion about the key problems solved in this paper, which is largely different from existing works. 

2. A deep literature review should be given, particularly regarding state-of-the-art deep learning methods for 3D LiDAR point clouds, e.g., 10.1109/CVPR46437.2021.01119, and 10.1016/j.isprsjprs.2023.03.022. 

3. Please clarify the contributions further. For example, which are your existing ones and which are your own ones?

4. It is well-known that the point data tend to suffer from various occlusions in the point cloud processing. Please give the discussion and analysis by referring to the paper titled e.g., ASFM-Net: Asymmetrical Siamese Feature Matching Network for Point Completion. The reviewer is wondering what will happen if the proposed method meets the different levels of occlusions.

5. The writing should be improved. The paper is not easy to read, for example, the caption of Table 1 should be corrected.

Comments on the Quality of English Language

1. Line 207, 'The rough set self-attention.' -> so strange there

2. Line 300, 'Part segmentation' -> 'part segmentation'

Author Response

Dear Reviewer,

I would like to express my sincere gratitude for your valuable feedback and comments on our entitled "RST: Rough set Transformer for 3D Point Cloud Learning" (Manuscript ID:sensors-2659157). Your insights have been instrumental in improving the quality and comprehensibility of our paper. Under your guidance, we have made necessary revisions and enhancements to address your recommendations. Revised portion are marked in red in the paper. Below, I provide responses to your suggestions and outline the changes we have implemented:

  1. Please give more details and discussion about the key problems solved in this paper, which is largely different from existing works.

We sincerely appreciate the valuable comments. This greatly contributes to the enhancement of the quality of our manuscript. This study primarily focuses on the problems in existing transformers that disregard the issues of confusion and misguidance arising from the disorderly and random nature of point clouds, especially within the context of global guidance.

In a detailed discussion, existing work utilizes an attention mechanism within the transformer framework that predominantly quantifies token relationships through dot products to guide the process. However, in cases where certain tokens convey incorrect information, this approach may result in a globally unobjective form of bootstrapping. The inherent randomness and disorder within point cloud data can readily give rise to tokens representing uncertainty. Therefore, the introduction of a novel feature representation method is essential to effectively handle uncertain tokens.

Building upon this, we propose an attention mechanism based on rough sets to incorporate tolerance for uncertain tokens during the guidance process. We employ neighborhood granulation to naturalize point cloud features to form information granules. The feature guidance is subsequently accomplished by measuring the similarity between information granules through lower approximation. Compared to the existing dot-product attention in transformers, this approach better tolerates the uncertainty inherent in point cloud data and provides a more objective representation. Consequently, it effectively compensates for the limitations of the current point cloud transformer.

Under the guidance of your comment, we have incorporated the aforementioned changes into the manuscript, enabling a comprehensive discussion of the key issues.

  1. A deep literature review should be given, particularly regarding state-of-the-art deep learning methods for 3D LiDAR point clouds, e.g., 10.1109/CVPR46437.2021.01119, and 10.1016/j.isprsjprs.2023.03.022.

We appreciate you very much for your constructive suggestion on our manuscript. Based on your suggestion, we conducted a more in-depth literature review. We believe that the discussion of "10.1109/CVPR46437.2021.01119" and "10.1016/j.isprsjprs.2023.03.022" significantly contributes to improving the quality of our manuscript, enabling readers to gain a more comprehensive understanding of prior research in this field. We extend our gratitude for your valuable professional suggestions, which are essential for enhancing the quality of our manuscript. We have incorporated additional references on deep learning methods for 3D LiDAR point clouds into the "Related Work" section of the revised manuscript.

  1. Please clarify the contributions further. For example, which are your existing ones and which are your own ones?

We appreciate your insightful feedback. The mathematical properties of rough set theory and the structural framework of the transformer are existing research works. Our main contributions are as follows:

  1. We redefine the granulation and lower approximation expressions for neighborhood rough set to conform to the fundamental definition of rough sets and enhance their applicability in deep learning.
  2. We propose a novel rough-set-based attention mechanism to replace the dot-product attention,thereby constructing a transformer network structure tailored for point cloud learning.

In line with your suggestions, we have restated our contributions at the end of the introduction for greater clarity.

  1. It is well-known that the point data tend to suffer from various occlusions in the point cloud processing. Please give the discussion and analysis by referring to the paper titled e.g., ASFM-Net: Asymmetrical Siamese Feature Matching Network for Point Completion. The reviewer is wondering what will happen if the proposed method meets the different levels of occlusions.

We greatly appreciate the value of your feedback to our manuscript. We agree that addressing occlusion-related challenges is a valuable research area. As the motivation for our article demonstrates, point cloud data contains various uncertainties that challenge transformer models, and occlusion is one of them. Thus, our aim is to introduce a novel guidance mechanism to address the transformer's limited capacity to handle these uncertain features.

In theory, we aggregate features into information granules using neighborhood relations and then use lower approximation to measure the similarity between these granules, enabling global guidance. Compared to traditional attention mechanisms, this approach demonstrates increased tolerance to uncertain features that may arise in situations like occlusion. To validate our capacity to manage uncertain features, we conducted experiments. In Figure 5, our experiments involved the recognition of point cloud data that was made incomplete and entirely random, illustrating RST's efficacy in handling incomplete information. This indirectly demonstrates our capability to address occlusion-related challenges.

Naturally, we believe your comment is highly relevant. As a result, we have referred to the ASFM-Net supplement you mentioned for comparative experiments involving various degrees of occlusion. We devised three levels (10%, 30%, 50%) of feature masking to assess the model's learning performance, and the results of the comparative experiments with the traditional transformer are presented below:

Figure 6. Comparison experiments between traditional transformer and RST with different occlusion levels.

Analyzing the experimental results, it becomes evident that as occlusion intensifies, the performance of the dot product attention-based Transformer significantly deteriorates, whereas RST displays robustness in handling occluded features. This is attributed to the rough set-based attention mechanism's ability to tolerate occluded tokens, resulting in more objective guidance.

This part of the experiment and its analysis will be added to the revised manuscript.

      5.The writing should be improved. The paper is not easy to read, for example, the caption of Table 1 should be corrected.

Thanks for your valuable suggestions about writing issues. We have conducted a thorough revision of the manuscript, addressing and rectifying all formatting issues in the revised version. For example, the title of Table 1 has been revised to "Comparison on the ModelNet40 classification dataset.". Additionally, we enlisted the assistance of an English major to polish our articles. We hope the revised manuscript could be acceptable for you.

For Comments on the Quality of English Language:

1.We have uniformly corrected 'The rough set self-attention.' to 'The rough set-based attention mechanism.'

2.Writing standards were re-examined and corrected.

We have conducted a comprehensive review of the entire manuscript in English to ensure that similar details have been rectified.

Once again, thank you very much for your comments and suggestions.

Reviewer 2 Report

Comments and Suggestions for Authors

1. The paper title needs to be slightly modified, when it comes to point cloud, it refers to 3D point cloud. So, RST: Rough set Transformer for Point Cloud Learning is better.

2. Minimize first person discourse in the content of the paper, like “We mainly compare” “We compare” “We can see that” …..

3. The abstract does not quite correspond to the core content of the paper. Please further refine the innovative points based on the content of the paper and display them in the abstract section.

4. In section 3, the content of 3.1 and 3.2 can be combined together. Meanwhile, this section can focus on highlighting the innovative points of one's own methods, rather than discussing existing methods.

5. In experiment section, overall accuracy (OA) and mean accuracy (mA) were compared in experiment 1, Can some efficiency comparative analysis be added in the experiment?

6. The paper mainly applies transfer learning of related model technology.  Is it suitable for different types of point clouds? Is it suitable for the segmentation of huge amounts of point clouds?

7. The segmentation effect in Experiment 2 is not very significant.  Can it be applied to actual object point cloud data for segmentation and visualization?

8. Suggest supplementing experimental data on different types of point cloud segmentation to demonstrate the universality of the paper's method.

Comments on the Quality of English Language

 Minor editing of English language required

Author Response

Dear Reviewer,

I would like to express my sincere gratitude for your valuable feedback and comments on our entitled "RST: Rough set Transformer for 3D Point Cloud Learning" (Manuscript ID:sensors-2659157). Your insights have been instrumental in improving the quality and comprehensibility of our paper. Under your guidance, we have made necessary revisions and enhancements to address your recommendations. Below, I provide responses to your suggestions and outline the changes we have implemented:

  1. The paper title needs to be slightly modified, when it comes to point cloud, it refers to 3D point cloud. So, RST: Rough set Transformer for Point Cloud Learning is better.

Thank you very much for your professional suggestions. We agree with your scientific expression, which has given a quality boost to our manuscripts. We have modified the title of the paper to “RST: Rough set Transformer for Point Cloud Learning”.

  1. Minimize first person discourse in the content of the paper, like “We mainly compare” “We compare” “We can see that” …

Thanks for your valuable suggestions about these writing issues. We reworked the writing of the paper to improve the quality. We make the necessary revisions to replace phrases like "We mainly compare," "We compare," "We can see that," and similar instances with more formal and objective language to enhance the overall quality of the paper. In particular, we have replaced "we compare most representative models" with "Results are compared to the most representative models," and "We can see that" has been replaced with "It is evident that," and so forth.

  1. The abstract does not quite correspond to the core content of the paper. Please further refine the innovative points based on the content of the paper and display them in the abstract section.

The comment you made has a very rigorous scientific spirit. Our innovations fall into two main areas: 

  1. We redefine the granulation and lower approximation expressions for neighborhood rough set to conform to the fundamental definition of rough sets and enhance their applicability in deep learning.
  2. We propose a novel rough-set-based attention mechanism to replace the dot-product attention, thereby constructing a transformer network structure tailored for point cloud learning.

Empirically, this represents the initial endeavor to combine rough set theory and transformer networks for point cloud learning.

We have reworked the abstract based on your suggestions. The details are as follows:

Point cloud data generated by LiDAR sensors plays a critical role in 3D sensing systems, with applications encompassing object classification, part segmentation, and point cloud recognition. Leveraging the global learning capacity of dot-product attention, transformer have recently exhibited outstanding performance in point cloud learning tasks. Nevertheless, existing transformer models inadequately address the challenges posed by uncertainty features in point clouds, which can introduce errors in the dot-product attention mechanism. In response to this, our study introduces a novel global guidance approach to tolerate uncertainty and provide more reliable guidance. We redefine the granulation and lower approximation operators based on neighborhood rough set theory. Furthermore, we introduce a rough set-based attention mechanism tailored for point cloud data and present the Rough Set Transformer (RST) network. Our approach utilizes granulation concepts derived from token clusters, enabling us to explore relationships between concepts from an approximation perspective, rather than relying on specific dot product functions. Empirically, our work represents the pioneering fusion of rough set theory and transformer networks for point cloud learning. Our experimental results, including point cloud classification and segmentation tasks, demonstrate the superior performance of our method. Our method establishes concepts based on granulation generated from clusters of tokens. Subsequently, we can explore relationships between concepts from an approximation perspective, instead of relying on specific dot product or addition functions. Empirically, our work represents the pioneering fusion of rough set theory and transformer networks for point cloud learning. Our experimental results, including point cloud classification and segmentation tasks, demonstrate the superior performance of our method. The resource codes are available at https://github.com/WinnieSunning/RST.

  1. In section 3, the content of 3.1 and 3.2 can be combined together. Meanwhile, this section can focus on highlighting the innovative points of one's own methods, rather than discussing existing methods.

We appreciate you very much for your constructive comments on our manuscript. We have relocated the discussion of existing work from Section 3 to the preceding section. The revised manuscript combines Sections 3.1 and 3.2, with a heightened focus on defining the rough-set attention mechanism and designing the overall structure of the Transformer network model. Specific changes to this section are reflected in the revised draft.

  1. In experiment section, overall accuracy (OA) and mean accuracy (mA) were compared in experiment 1, Can some efficiency comparative analysis be added in the experiment?

The comment you made has a very rigorous scientific spirit. Efficiency is a topic of concern. To this end, we add a comparative experiment between the traditional transformer model and the rough set-based transformer in terms of efficiency. Here are our time tests on the RTX2060 with a batchsize of 16:

Table 4. Comparison experiments between traditional Transformer and RST with efficiency

Method Epochs(convergence) Average training time Average testing time
Traditional transformer 250 79.13s/epoch 8.58s
RST(ours) 300 79.58s/epoch 8.67s

Clearly, the rough set-based transformer is more time-consuming than the traditional transformer(dot-product attention) due to its intricate algorithm design and additional computations. Nevertheless, we have optimized it to reduce the time required, incorporating algorithm enhancements and developing CUDA parallel computing libraries. At present, the time required by our method is nearly equivalent to that of the traditional transformer. We have open-sourced the code and library files for this approach as part of our contribution. As a result, our approach offers a more comprehensive performance improvement, leading to superior overall efficiency compared to the traditional transformer.

The above experiments and discussions will also be added to the revised manuscript.

  1. The paper mainly applies transfer learning of related model technology.Is it suitable for different types of point clouds? Is it suitable for the segmentation of huge amounts of point clouds?

Thank you very much for commenting on the sentence of our manuscript. Our approach demonstrates strong generalization capabilities, both in theory and through experimental evidence presented in the paper. In our experiments, we classify 40 sets of point cloud data in various forms. In Figure 5, we randomly input a set of point cloud data with 40 different class labels to assess our model's recognition capabilities. The results confirm the applicability of our method to diverse types of point clouds. Furthermore, we conducted experiments to assess generalizability in other domains, including the cifar100, CMU-MOSEI, and MELD datasets, all of which demonstrated strong performance. Among them, our method attains an 85% F1-score on the IEMOCAP dataset for multimodal sentiment analysis, achieving state-of-the-art performance. These results serve as strong evidence of our model's robust generalizability to various data inputs. Of course, we acknowledge the importance of supplementing additional types of point cloud data, and as such, this will be a focal area of our future research endeavors.

Regarding segmentation experiments with substantial amounts of point cloud data, we utilized the ShapeNet dataset comprising 16 major classes and 50 minor classes. The sufficiently sized point cloud data we use as input in this dataset can to some extent reflect our ability to learn from large-scale point clouds. Since we want to emphasize the innovativeness of the first combined application of rough set theory and transformer for point cloud learning, more challenging datasets will be used to evaluate in future work, such as the ScanNet and PartNet.

  1. The segmentation effect in Experiment 2 is not very significant.  Can it be applied to actual object point cloud data for segmentation and visualization?

We appreciate the reviewer's attention to and concern regarding the segmentation results in Experiment 2. In response, we acknowledge that the segmentation experiment results may not be particularly significant, as our primary aim is to underscore the effectiveness of the initial combined application of rough set theory and transformer to point cloud learning. Concerning the application of our method to real-world object point cloud data for segmentation and visualization, several articles have performed validations, including "PVT: Point-Voxel Transformer for Point Cloud Learning" and "PatchFormer: An Efficient Point Transformer with Patch Attention." Given that our experimental setup closely mirrors theirs to ensure comparative efficacy and to surpass prior works in terms of performance, our method is indeed suitable for real-world object point cloud data segmentation and visualization.

Naturally, we find your comments quite meaningful. To further validate the model's real-world applicability, we have added comparison experiments with simulated realistic occlusions, and the results are presented below:

Figure 6. Comparison experiments between traditional transformer and RST with different occlusion levels.

Analyzing the experimental results, it becomes evident that as occlusion intensifies, the performance of the dot product attention-based transformer significantly deteriorates, whereas RST displays robustness in handling occluded features. This is attributed to the rough set-based attention mechanism's ability to tolerate occluded tokens, resulting in more objective guidance. This part of the experiment and its analysis will be added to the revised manuscript.

Absolutely, the proposed method's applicability to actual object point cloud data for segmentation and visualization is certainly a valuable direction for future research. We will incorporate this point into our revised manuscript, recognizing the necessity for additional research and validation of our approach using real-world point cloud data to evaluate its practical usefulness in segmentation and visualization tasks. We will place strong emphasis on this as a future research direction to ensure a more thorough assessment of our method's applicability and effectiveness. We appreciate your valuable suggestion.

     8.Suggest supplementing experimental data on different types of point cloud segmentation to demonstrate the universality of the paper's method.

We appreciate your insightful suggestion. In response to this valuable input, we concur that showcasing the universality and versatility of our method is essential to enhance the comprehensiveness of our paper. We recognize the significance of confirming universality, and we have already validated our approach in domains such as image classification, named-entity recognition, and multimodal sentiment analysis, all of which demonstrated outstanding performance. Consequently, we can ensure the model's universality. Due to constraints related to experimental equipment and conditions, we will emphasize different types of point cloud segmentation experiments in future work. We will include a variety of datasets, including ScanNet and PartNet.

For Comments on the Quality of English Language:

We have conducted a comprehensive editing and proofreading of the entire manuscript to ensure the quality of the paper.

Once again, thank you very much for your comments and suggestions.

Round 2

Reviewer 1 Report

Comments and Suggestions for Authors

My concerns have been addressed, so I tend to accept this paper.